# Impact of commonly used drugs on the composition and metabolic function of the gut microbiota

Arnau Vich Vila [1,2], Valerie Collij[1,2], Serena Sanna [2], Trishla Sinha[2], Floris Imhann[1,2], Arno R. Bourgonje[1], Zlatan Mujagic[3], Daisy M.A.E. Jonkers[3], Ad A.M. Masclee[3], Jingyuan Fu[2], Alexander Kurilshikov[2], Cisca Wijmenga [2], Alexandra Zhernakova[2] & Rinse K. Weersma[1]*

The human gut microbiota has now been associated with drug responses and efficacy, while chemical compounds present in these drugs can also impact the gut bacteria. However, drug–microbe interactions are still understudied in the clinical context, where polypharmacy and comorbidities co-occur. Here, we report relations between commonly used drugs and the gut microbiome. We performed metagenomics sequencing of faecal samples from a population cohort and two gastrointestinal disease cohorts. Differences between users and non-users were analysed per cohort, followed by a meta-analysis. While 19 of 41 drugs are found to be associated with microbial features, when controlling for the use of multiple medications, proton-pump inhibitors, metformin, antibiotics and laxatives show the strongest associations with the microbiome. We here provide evidence for extensive changes in taxonomy, metabolic potential and resistome in relation to commonly used drugs. This paves the way for future studies and has implications for current microbiome studies by demonstrating the need to correct for multiple drug use.

[1] Dept. of Gastroenterology and Hepatology, University of Groningen and University Medical Center Groningen, Groningen, The Netherlands. [2] Dept. of Genetics, University of Groningen and University Medical Center Groningen, Groningen, The Netherlands. [3] Division Gastroenterology Hepatology, Maastricht University Medical Center, Maastricht, The Netherlands. *email: r.k.weersma@umcg.nl

n recent years there has been growing interest in the associations between the gut microbial ecosystem and the use of non-antibiotic drugs. The interaction between drugs and gut microbe composition is important for understanding drug mechanisms and the development of certain drug side effects[1,2]. The impact of antibiotics on gut microbiome composition has been well known for some time, but studies in population-based cohorts have found relations between multiple groups of drugs and gut microbiome signatures[3–6]. The use of proton pump inhibitors (PPIs), drugs that inhibit stomach acid production, has been associated with an increase in typically oral bacteria in the gut[7,8]. Metformin, a commonly used drug in type II diabetes, has been associated with changes in gut microbiome composition both in vivo and in mice, in particular with an increase in bacteria that produce short chain fatty acids[9,10]. A recent study in a general population cohort showed that multiple drugs are associated with an altered gut microbiome composition[6]. In the same line, in vitro analysis of more than 1000 marketed drugs showed that non-antibiotic drugs can also inhibit the growth of gut bacterial strains[11]. This, together with the fact that gut microbial composition has been linked to host conditions such as rheumatoid arthritis, inflammatory bowel disease (IBD) and susceptibility to enteric infections, suggests that some drug side effects could be induced via their impact on the gut ecosystem[7,12–15]. To date, most of the studies published on this topic have focused on general population cohorts or single drug–microbe interactions[4–6]. However, these approaches do not reflect the clinical situation. Patients with gastrointestinal (GI) diseases like IBD and irritable bowel syndrome (IBS), for example, harbour a different gut microbiota composition than general population controls[15], and this could influence the occurrence of side effects or alter the mechanism of action of certain drugs. Moreover, patients with IBD or IBS also show differences in their patterns of drug use compared to general population controls, including increased polypharmacy, either due to the GI disease itself or to other comorbidities[16–19]. In IBS, many commonly used drugs such as nonsteroidal anti-inflammatory drugs (NSAIDs) or antidepressants can trigger or alleviate GI symptoms[19]. Investigating drug–microbiome interactions could therefore lead to insights that can unravel the mechanisms involved in treatment response in IBD and the occurrence of GI symptoms with drug use in IBS. To understand the impact of drug–microbiome interaction in humans, especially in the clinical context, it is crucial to consider the combination of different drug types.

Here we present a meta-analysis of the associations between drug use and the gut microbiome in three independent cohorts from the same geographical region. After correcting for the age, sex and BMI of the participants, 19 of the 41 medication categories available in this study show an association with microbial features. When we also correct for the use of multiple drugs at the same time, PPI, metformin, antibiotics and laxatives show the largest number of associations. Through this data, we pinpoint relevant changes in microbial species and metabolic pathways and consequences for antibiotic resistance (AR) mechanisms in the gut in clinical context.

## Results

**Drug use.** In this study, we used three Dutch cohorts: a general population cohort, a cohort of patients with IBD and a case-control cohort of patients with IBS (see methods). 1126 of the 1883 participants from all three cohorts were taking at least one drug at time of faecal sampling. The number of drugs used per participant ranged from 0 to 12, with median values of 0 for the population cohort (mean 1.03, $n = 1124$), 2 for the IBD cohort

(mean 2.35, $n = 454$) and 1 for the IBS cohort (mean 1.6, $n = 305$) (Table 1, Supplementary Data 1 and 2). In total, we observed 537 different combinations of drugs, with the most frequent being the combination of a beta-sympathomimetic inhaler with a steroid inhaler (18 users) (Supplementary Data 2). The use of steroid inhalers was strongly correlated with the use of beta sympathomimetic inhalers ($R_{population-cohort} = 0.78$, $R_{IBD-cohort} = 0.65$, $R_{IBS-cohort} = 0.78$, Spearman correlation, False Discovery Rate [FDR] < 0.05) (Supplementary Data 3–5). In patients with IBD, the strongest correlation was observed between calcium and

| Drugs | LifeLinesDEEP (n = 1124) | 1000 IBD (n = 454) | MIBS (n = 305) |
|---|---|---|---|
| ACE inhibitors | 44 (4%) | 24 (5%) | 7 (2%) |
| Alpha blockers | 10 (1%) | 3 (1%) | 7 (2%) |
| AngII receptor antagonist | 33 (3%) | 10 (3%) | 17 (6%) |
| Anti-androgen oral contraceptive | 14 (1%) | 2 (0%) | 6 (2%) |
| Anti-epileptics | 5 (0%) | 5 (1%) | 7 (2%) |
| Antihistamine | 69 (6%) | 15 (4%) | 14 (5%) |
| Anti-TNFα | 1 (0%) | 119 (25%) | 0 (0%) |
| Antibiotics merged | 13 (1%) | 12 (3%) | 7 (3%) |
| Benzodiazepine derivatives related | 25 (2%) | 16 (4%) | 13 (5%) |
| Beta blockers | 61 (5%) | 34 (8%) | 23 (8%) |
| Beta sympathomimetic inhaler | 64 (6%) | 16 (4%) | 16 (6%) |
| Bisphosphonates | 10 (1%) | 13 (3%) | 4 (1%) |
| Ca-channel blocker | 21 (2%) | 10 (2%) | 14 (5%) |
| Calcium | 14 (1%) | 76 (17%) | 8 (3%) |
| Iron preparations | 7 (1%) | 15 (3%) | 1 (0%) |
| Folic acid | 7 (1%) | 31 (7%) | 0 (0%) |
| Insulin | 4 (0%) | 11 (2%) | 0 (0%) |
| IUD that includes hormones | 60 (5%) | 5 (1%) | 1 (0%) |
| K-saving diuretic | 7 (1%) | 9 (2%) | 1 (0%) |
| Laxatives | 21 (2%) | 30 (7%) | 27 (9%) |
| Levothyroxine | 26 (2%) | 10 (2%) | 5 (2%) |
| Melatonin | 6 (1%) | 4 (1%) | 1 (0%) |
| Mesalazines | 2 (0%) | 162 (36%) | 2 (1%) |
| Metformin | 15 (1%) | 7 (2%) | 6 (2%) |
| Methylphenidate | 6 (1%) | 5 (1%) | 1 (0%) |
| NSAID | 42 (4%) | 21 (5%) | 22 (7%) |
| Opiate | 13 (1%) | 22 (5%) | 7 (2%) |
| Oral anti-diabetics | 8 (1%) | 8 (2%) | 4 (1%) |
| Oral contraceptive | 113 (10%) | 55 (12%) | 32 (11%) |
| Oral steroid | 5 (0%) | 79 (17%) | 4 (1%) |
| Other antidepressant | 9 (1%) | 10 (2%) | 3 (1%) |
| Paracetamol (acetaminophen) | 11 (1%) | 42 (9%) | 42 (14%) |
| Platelet aggregation inhibitor | 32 (3%) | 27 (6%) | 18 (6%) |
| PPI | 93 (8%) | 108 (24%) | 48 (16%) |
| SSRI antidepressant | 28 (2%) | 10 (2%) | 30 (10%) |
| Statin | 55 (5%) | 28 (6%) | 26 (9%) |
| Steroid inhaler | 57 (5%) | 17 (4%) | 17 (6%) |
| Steroid nose spray | 55 (5%) | 6 (1%) | 7 (2%) |
| Thiazide diuretic | 43 (4%) | 17 (4%) | 17 (6%) |
| Thiopurines | 0 (0%) | 151 (33%) | 0 (0%) |
| Tricyclic antidepressant | 10 (1%) | 16 (4%) | 2 (1%) |
| Triptans | 20 (2%) | 5 (1%) | 2 (1%) |
| Vitamin D | 14 (1%) | 70 (15%) | 3 (1%) |
| Vitamin K antagonist | 5 (0%) | 7 (2%) | 6 (2%) |

Table 1 Drug usage per cohort. Number and percentage of drug users in each cohort.

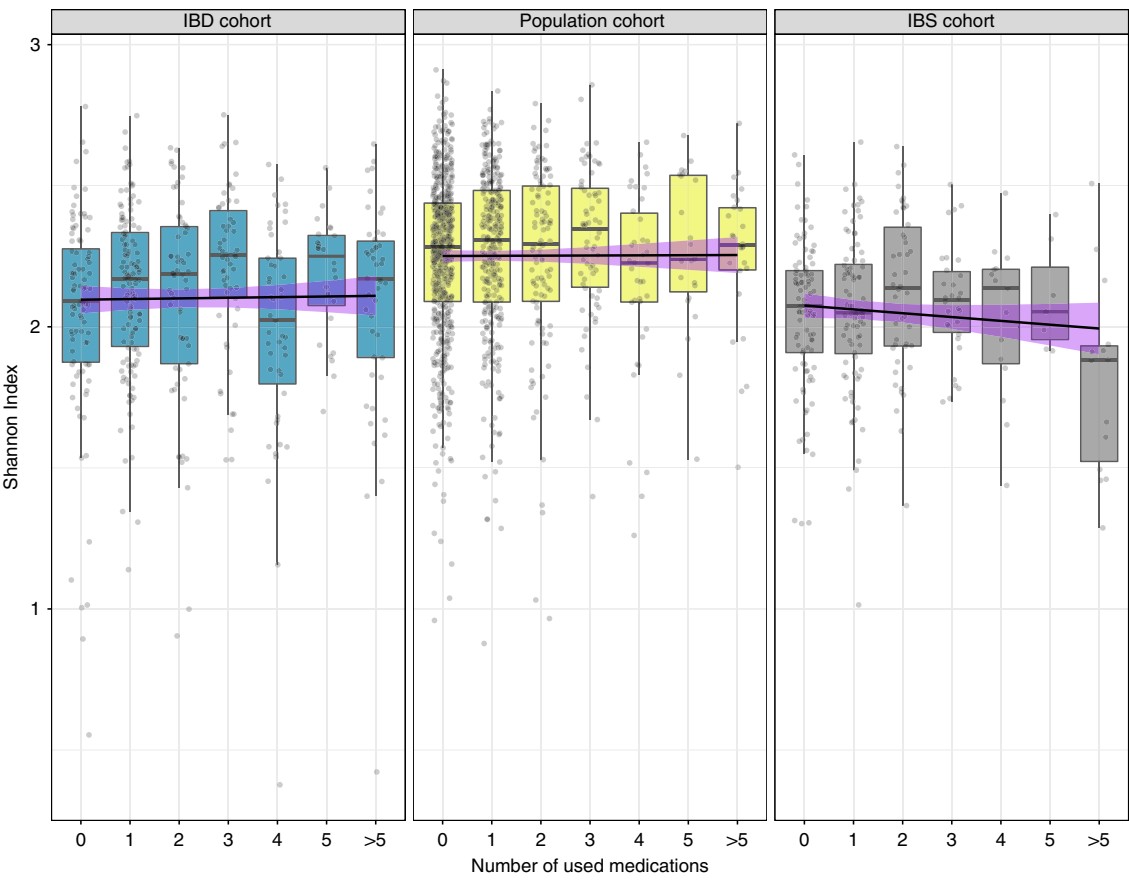

**Fig. 1 Microbial richness (Shannon index) for each participant stratified per number of medications used.** Dots represent the richness value per each sample in the study. Boxplot shows the median and interquartile range (25th and 75th). Whiskers show the 1.5*IQR range. Black lines show linear regression with a purple shadow indicating the 95% confidence interval. From left to right, the IBD cohort ($N = 454$ samples, linear regression, coefficient $= -0,001$, $p = 0.88$), population cohort ($N = 1124$ samples, linear regression, coefficient $= -0.002$, $p = 0.77$) and the IBS cohort ($N = 305$ samples, linear regression, coefficient $= -0.016$, $p = 0.06$). (Source data are provided as a Source Data file).

vitamin D supplements (R = 0.84, Spearman correlation, FDR < $2 \times 10^{-16}$). Mesalazines (36%), thiopurines (33%) and anti-TNFα agents (25%) were present in the top 10 most-used drugs in the IBD cohort (Table 1). Since thiopurines and anti-TNFα agents were solely used in the IBD cohort, these drugs were not included in our multi-drug analyses.

**Microbial ecosystem and drug use**. We first investigated the effect of each individual drug on the richness and overall gut microbial composition. As described earlier, disease cohorts presented a lower microbial richness compared to the general population cohort (Population cohort Shannon Index$_{mean}$ = 2.26 (0.96–2.91), IBD cohort Shannon Index$_{mean}$ = 2.1 (0.38–2.78), IBS cohort Shannon Index$_{mean}$ = 2.02 (1.01–2.65))[15]. Within cohorts, we did not observe any significant changes in the microbial richness associated with the use of any drug or in the number of different drugs used (Spearman correlation, FDR > 0.05, Supplementary Data 6, Fig. 1). However, we did observe differences between the number of drugs used and the overall microbial composition within all cohorts (Permutational multivariate analysis of variance (PERMANOVA) test; Population cohort: $r^2 = 0.006$, FDR = 0.001; IBD cohort: $r^2 = 0.015$, FDR = 0.001; IBS cohort: $r^2 = 0.014$, FDR = 0.0014; Supplementary Data 7). PPIs were the only individual drug associated with compositional changes in all cohorts (PERMANOVA, Population cohort: $r^2 = 0.007$, FDR = 0.006; IBD cohort: $r^2 = 0.023$, FDR = 0.0006; IBS cohort: $r^2 = 0.021$, FDR = 0.01).

**Taxa and pathways associated with drug use**. In the meta-analysis accounting for host age, sex, BMI and sequencing depth, 154 associations between individual taxa and 17 groups of drugs were found to be statistically significant (inverse variance meta-analysis, FDR < 0.05, Fig. 2a, Supplementary Data 8). PPIs, met-formin, vitamin D supplements and laxatives were the individual drugs with the highest number of associations (>10) in the single-drug analysis. An interesting observation was that changes in the abundance of specific taxa were associated with multiple independent drugs. For example, the abundance of *Streptococcus salivarius* was increased in users of opiates, oral steroids, platelet aggregation inhibitors, PPIs, SSRI antidepressants and vitamin D supplements (inverse variance meta-analysis, FDR < 0.05). We did, however, also see features that were specific to individual drugs: an increased abundance of *Bifidobacterium dentium* was specific to PPI users (inverse variance meta-analysis, FDR = $9.38 \times 10^{-17}$) and an increased abundance of *Eubacterium ramulus* was specific to participants using SSRI antidepressants (inverse variance meta-analysis, FDR = 0.047). The use of drugs was also associated with functional changes in the gut. In the same analysis, 411 microbial pathways were related to 11 drugs (inverse variance meta-analysis, FDR < 0.05, Fig. 2b, Supplementary Data 9).

In order to consider multiple drug groups being prescribed at the same time, we tested each drug adding other drugs as covariates in the linear models. Overall, 47 associations were found between microbial relative abundance and the use of six drugs (inverse variance meta-analysis, FDR < 0.05, Fig. 1b,

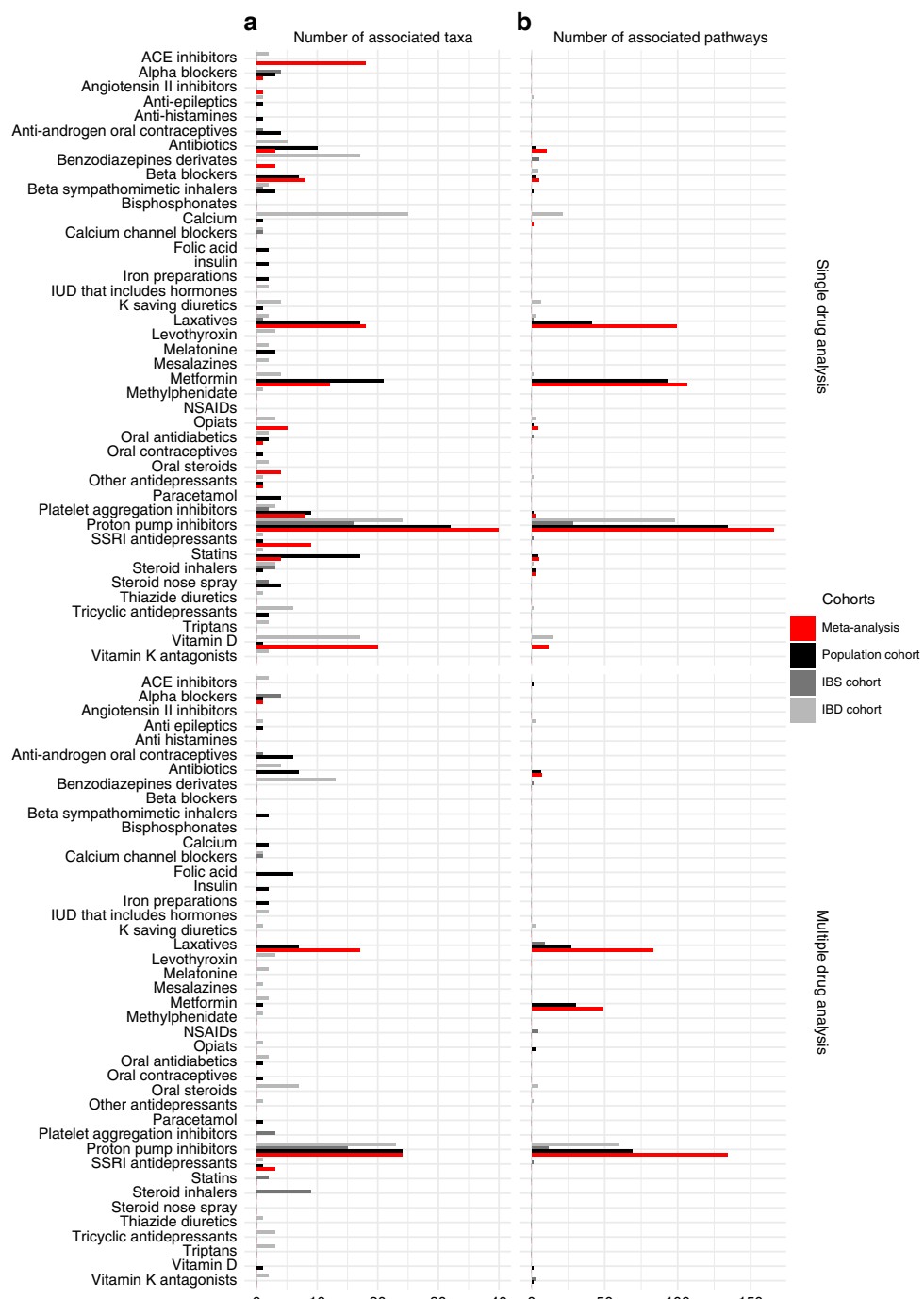

**Fig. 2 Overview of the number of associated microbial features. a,b** Bar-plots showing the number of associations between each type of drug and microbial taxa **a** and microbial pathways **b**. Bar colours indicate the results from the separate cohorts and the results from the meta-analyses. Black bars indicate the population cohort, dark grey bars the IBS cohort, light grey bars the IBD cohort and red bars the results from the meta-analyses. The single-drug model shows the association when considering one drug at the time while taking age, sex and BMI into account. The multi-drug model considers the use of multiple drug types while taking age, sex and BMI into account. (Source data are provided as a Source Data file).

Supplementary Data 10). PPIs, laxatives and metformin showed the largest number of associations with microbial taxonomies and pathways. Despite the low number of antibiotics users, a decrease in the genus *Bifidobacterium* was observed in this meta-analysis (inverse variance meta-analysis, FDR = 0.001). Laxative users showed higher abundance of *Alistipes* and *Bacteroides* species in their microbiome (inverse variance meta-analysis, FDR < 0.05). The association between SSRI antidepressant use and *Eubacterium ramulus* remained significant after considering multi-drug

use (inverse variance meta-analysis, FDR = 0.032). In this multi-drug analysis, 271 pathways were associated with four drug categories: PPIs, laxatives, antibiotics and metformin (inverse variance meta-analysis, FDR < 0.05, Supplementary Data 11). Interestingly, while antibiotic use was related to a lower abundance of microbial pathways such as amino-acid biosynthesis, metformin use was associated with increased bacterial metabolic potential (inverse variance meta-analysis, FDR < 0.05). Within the category of antibiotics, tetracyclines showed the

strongest association with the altered pathways (two-sided Wilcoxon test, FDR < 0.05, Supplementary Data 12). Moreover, the abundance of microbial pathways in laxative- and PPI-users did show some similarities, including increase of glucose usage (increase of glycolysis pathways) and decrease of starch degradation and aromatic compounds biosynthesis pathways (inverse variance meta-analysis, FDR < 0.05). All the associations between taxonomy and pathways and individual drugs can be found in Supplementary Data 13–46.

**Cohort-specific changes in gut microbiome composition.** Changes in the gut microbiota composition in patients with IBD and IBS have been observed before[15,20,21]. We therefore examined whether these changes were also present in the associations between the microbiota and the use of a drug. In the IBD cohort, benzodiazepine use was associated with an increased abundance of *Haemophilus parainfluenzae* (linear regression, t-test, FDR = 0.008, Supplementary Data 47). Interestingly, this bacterium has also been described to be more prevalent in patients with IBD than in healthy individuals. The use of tricyclic antidepressants was associated with an increased abundance of *Clostridium leptum* and intake of levothyroxine was associated with a higher abundance of *Actinomyces* in the IBD cohort (linear regression, t-test, FDR = 0.005 and 0.005, respectively, Supplementary Data 47). In addition, the 17 steroid inhaler users in the IBS cohort showed a higher abundance of *Streptococcus mutans* and *Bifidobacterium dentium* in their gut microbiome (linear regression, t-test FDR = 0.001 and 0.01, respectively, Supplementary Data 40). Interestingly, patients with IBD using oral steroids showed a higher abundance of *Methanobrevibacter smithii* (linear regression, t-test, FDR = 0.004, Supplementary Data 47). This association was also reflected at pathway level: the four pathways associated with the use of this drug also showed a high correlation with the abundance of *Methanobrevibacter smithii* (Spearman correlation, rho > 0.93, FDR < 2 × 10⁻¹⁶, Supplementary Data 48 and Fig. 3). Two of these pathways are

involved in methanogenesis, one in the biosynthesis of vitamin B2 and one in the biosynthesis of nucleosides. Conversely, the use of other medication usually prescribed to treat IBD did not show strong associations with the microbial composition. Only the abundance of an *Erysipelotrichaceae* species was found to be slightly increased in mesalazine users (linear regression, t-test, FDR = 0.047, Supplementary Data 49).

**Microbiome signature of PPI users.** PPIs accounted for the largest number of associations, with a total of 40 altered taxa and 166 altered microbial pathways in the single-drug analyses (inverse variance meta-analysis, FDR < 0.05, Supplementary Data 8 and 9).

When correcting for the impact of other drug types, 24 taxa and 133 microbial pathways remained significantly associated with PPIs (inverse variance meta-analysis, FDR < 0.05). We observed an increased abundance of *Veillonella parvula*, which is known to establish a mutualistic relation with *Streptococcus mutans* by co-aggregating and transforming the metabolic products of carbohydrate-fermenting bacteria[22] (inverse variance meta-analysis, FDR = 1.61 × 10⁻⁶ and 6.13 × 10⁻²⁴, respectively).

Functional changes included the increase of fatty acid and lipid biosynthesis, fermentation NAD metabolism and biosynthesis of L-arginine. The pathways associated with PPI use involve functions that have a broad taxonomic contribution. However, a closer look at the predicted microbial contribution and the gene families involved in each pathway revealed that the enrichment of specific microbial mechanisms is likely to be explained by the changes observed in taxonomic composition. Purine deoxyribonucleoside degradation, a pathway used as a source of energy and carbon, was predicted from the genomes of >25 different bacterial genera (Fig. 4). The increase in this function in the gut microbiome of PPI users can be explained by an increased abundance of Streptococcus species (*S. salivaris, S. parasanguinis* and *S. vestibularis*) (two-sided Wilcoxon-test, FDR < 0.05). Three pathways involved in L-arginine biosynthesis (MetaCyc ID: PWY-7400, ARGSYNBSUB and ARGSYN) were more abundant

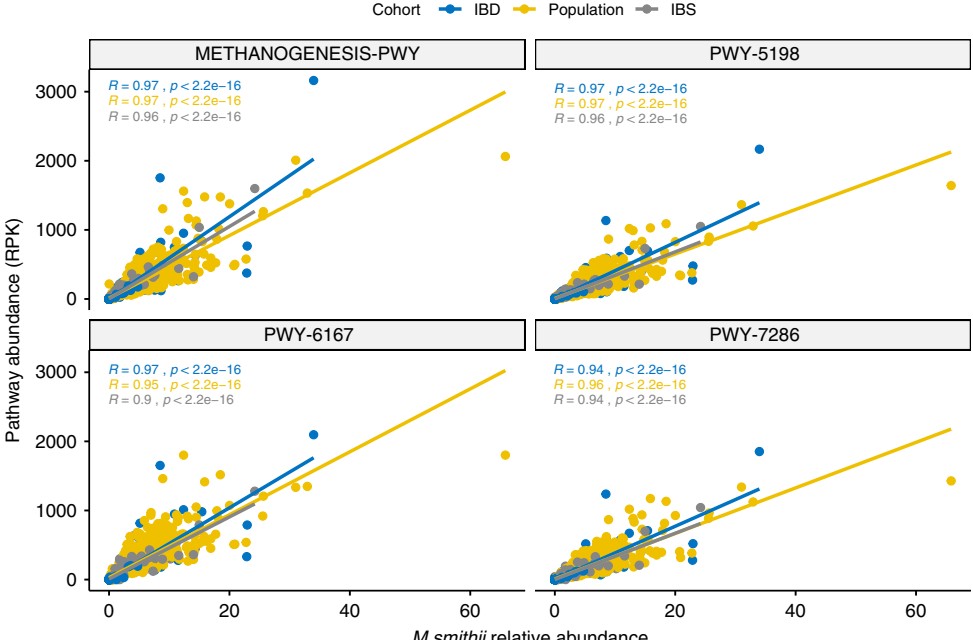

**Fig. 3 Correlation between the relative abundance of *Methanobrevibacter smithii* and the pathways associated with oral steroids.** Dots are coloured by study cohort. In blue the cohort of IBD patients, in yellow the population cohort and in grey the IBS case-control cohort. X-axis represents the relative abundance of *Methanobrevibacter smithii* and Y-axis the read per kilobase (RPK) of each pathway. Spearman correlation was used to calculate the correlation and significance. (Source data are provided as a Source Data file).

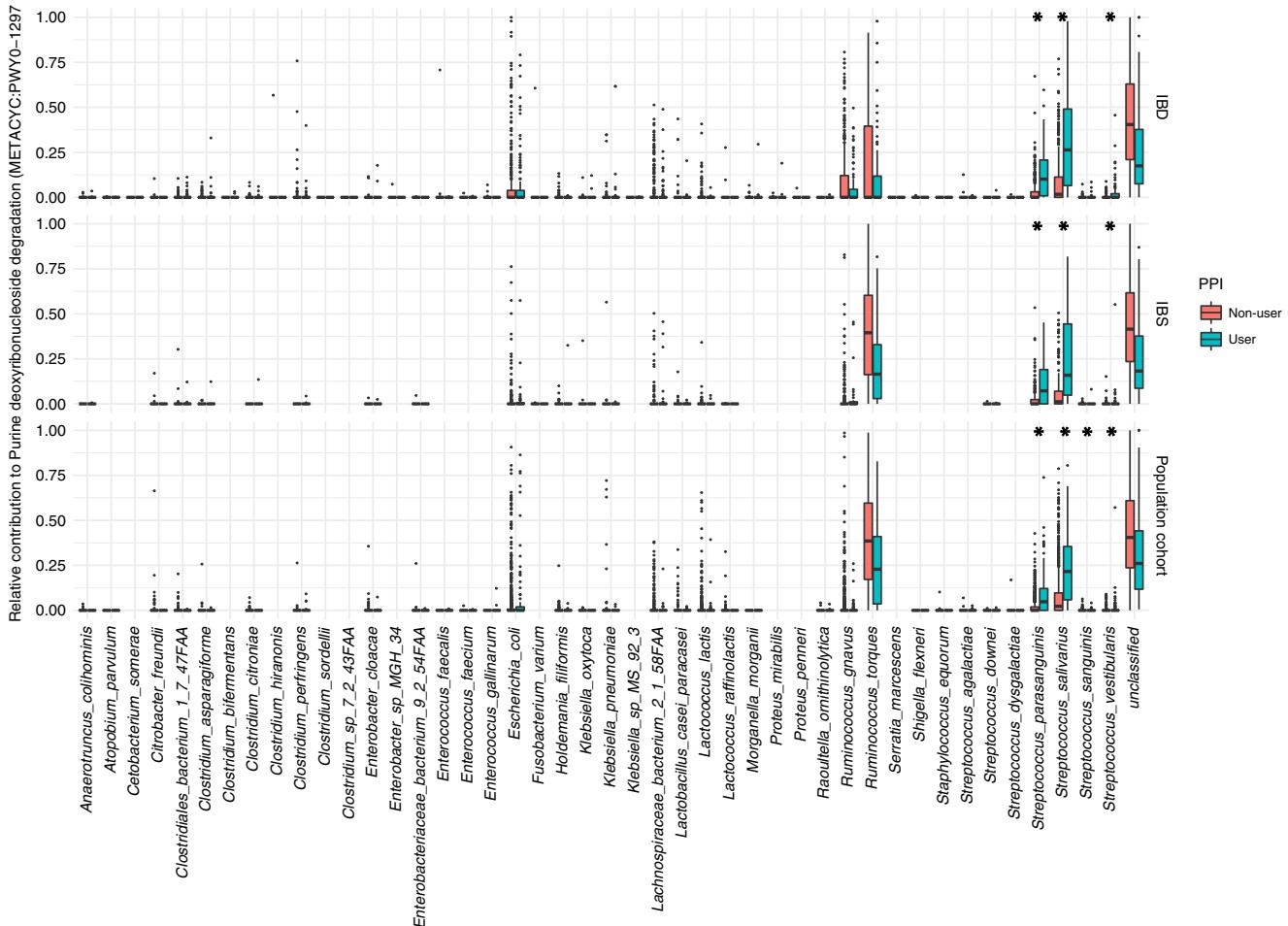

**Fig. 4 Microbial contribution to the purine deoxyribonucleoside degradation pathway.** Box plots represent the relative contribution of each microbe to the overall pathway quantification for each cohort separately. On top is the IBD cohort represented, in the middle the IBS cohort and on the bottom row the LLD cohort. Blue box-plots represent the values of PPI users. Red box-plots represent the values of non-PPI users. Asterisks indicate statistically significant differences between PPI users and non-users (Wilcoxon test, FDR < 0.05). Box plots show medians and the first and third quartiles (the 25th and 75th percentiles), respectively. The upper and lower whiskers extend the largest and smallest value no further than 1.5*IQR, respectively. Outliers are plotted individually.

in the microbiome of PPI users. While several bacterial taxa, including *Bifidobacterium* and *Ruminococcus* species were predicted to contribute to these pathways, only *Streptococcus mutans* pathways showed a significant enrichment (two-sided Wilcoxon-test, FDR < 0.05, Supplementary Data 50). These analyses have also been performed in the medication's metformin, antibiotics and laxatives (Supplementary Data 51–53).

Different types of PPIs, namely omeprazole, esomeprazole and pantoprazole, exhibited a similar effect on the gut microbiome. Additionally, of the 133 microbial pathways associated with PPI use, 46 pathways also showed dosage dependent effects (two-sided Wilcoxon test, FDR < 0.05, Supplementary Data 12). For example, participants using a higher dosage of PPIs (greater than or equal to 40 mg/day) showed a marked decrease in a pathway involved in the biosynthesis of amino acids (PWY-724) when compared to low dosage users (two-sided Wilcoxon test, FDR = 0.00065, Supplementary Data 12).

**Metformin is associated with changes in metabolic potential.** While changes in the abundance of *Streptococcus*, *Coprococcus* and *Escherichia* species were initially found to be enriched in metformin users, these associations were no longer significant

after correcting for the use of other drug types. However, a suggestive association with *Escherichia coli* (linear regression, *t*-test, FDR = 0.08, Supplementary Data 10) remained and, in the IBD cohort, the abundance of *Streptococcus mutans* was slightly increased in participants using this drug (linear regression, *t*-test, FDR = 0.05) (Supplementary Data 47).

Strikingly, the functional implications of metformin use were large even after correction for the use of other drugs, with 48 microbial pathways altered compared to the non-users (inverse variance meta-analysis, FDR < 0.05, Supplementary Data 11). Metformin use was associated with changes in the metabolic potential of the microbiome, in particular with increases in the butanoate production, quinone biosynthesis, sugar derivatives degradation and polymyxin resistance pathways (Supplementary Data 11). Interestingly, metagenomic pathway prediction and gene family analyses revealed that Enterobacteriaceae species, mainly *Escherichia coli*, were the major contributors to the functional changes associated with metformin use. Our data suggest that physiological changes induced by metformin can provide competitive advantage to enterobacterial species, which could potentially have implications on health (two-sided Wilcoxon-test, FDR < 0.05, Supplementary Data 51). Furthermore, we did not identify dosage-dependent effects of metformin

usage on the associated pathways (two-sided Wilcoxon test, FDR > 0.05, Supplementary Data 12).

**Effect of SSRI antidepressant use in patients with IBS.** SSRI antidepressants were among the top 10 most-used drugs in the IBS cohort. In SSRI users, the only taxa that remained statistically significant in the multi-drug meta-analysis was the increased abundance of *Eubacterium ramulus* (inverse variance meta-analysis, FDR = 0.032, Supplementary Data 10). This medication category included six different subtypes of drugs in which paroxetine represented 32% of the SSRI users (Supplementary Data 1). Interestingly, and despite the low numbers, the increased abundance of *Eubacterium ramulus* was mainly observed in the paroxetine users (two-sided Wilcoxon test, FDR = 0.054, P-value = 0.003, Supplementary Data 12).

Furthermore, the pathway involved in peptidoglycan maturation was decreased in the multi-drug meta-analysis of SSRI antidepressant users compared to non-users (inverse variance meta-analysis, FDR = 0.13). However, this finding was mainly observed in the IBS cohort (linear regression, t-test, FDR = 0.002, Supplementary Data 38).

**Drug use is associated with different resistome profiles.** In vitro evidence is becoming available that indicates it is not only antibiotic use that can increase AR: non-antibiotic drugs can also contribute to this mechanism[11]. To address this we first analysed for all three cohorts separately whether the total count of AR genes was increased in users of individual drugs compared to those who were not using any drugs. For the general population cohort, the total number of AR genes was increased for users of metformin and PPIs (two-sided Wilcoxon test, FDR = 0.04 and 0.04, respectively, Supplementary Data 54). In the IBS cohort, after correction for multiple testing, none of the drugs were associated with an increase in AR genes (two-sided Wilcoxon test, FDR > 0.05). In the IBD cohort, this increase was present in opiate and tricyclic antidepressant users (two-sided Wilcoxon-test, FDR = 0.04 and 0.04, respectively). If we use a less stringent FDR cutoff of < 0.25, 15 drugs were associated with the number of AR markers across the three cohorts. To identify which drug groups were related to an increase in individual AR genes, we analysed the abundances of individual AR genes. In all three cohorts, we identified consistent increases in three AR gene markers in PPI users compared to participants not using any drugs (two-sided Wilcoxon test, FDR < 0.05, Supplementary Data 55). These genes belong to *tetA*, *tetB* and *Mel* (ARO:3004033, ARO:3004032 and ARO:3000616, respectively), which are parts of efflux pumps that pump certain types of antibiotics out of the bacteria and thereby inhibit the functional mechanisms of these antibiotics[23,24]. For *tetA* and *tetB*, this affects the antibiotic group tetracyclines[23]. For *Mel*, it affects the antibiotic group macrolides[24].

These AR markers have the highest correlations with *Streptococcus parasanguinis* (Spearman correlation, rho's between 0.56–0.75, FDR < 0.05). We identified three ARs that have also been tested in vitro[11]. The AR *TolC*, for example, is known to be involved in multiple multi-drug efflux pumps[23] and was statistically significantly increased in six drug groups: three in our general population cohort (PPI, statin and metformin), two in our IBS cohort (steroid nose spray and levothyroxin) and one in our IBD cohort (tricyclic antidepressants) (two-sided Wilcoxon test, FDR < 0.05). Another example of an in vitro-tested AR is *mdtP*, another multi-drug resistance efflux pump, which was increased in metformin users in our general population cohort (two-sided Wilcoxon test, FDR = 0.001) and in tricyclic

antidepressant users in the IBD cohort (two-sided Wilcoxon test, FDR = 0.017, Supplementary Data 55).

## Discussion

In this study we have shown the influence of commonly used drugs on gut microbiome composition, microbial functions and AR mechanisms in both the general population and individuals with GI diseases while also considering the clinical context, where polypharmacy and comorbidities play an important role. We also observed how drug-associated changes have implications for the clinically relevant feature of AR. Interestingly, for 15 different drugs and across all three cohorts, we observed an increase in total AR genes in drug users compared to participants not using any drugs (FDR < 0.25).

We observed that the overall composition of the gut ecosystem is only consistently altered by the use of PPIs and by the use of multiple drugs. While the effect of PPI use can potentially be explained by changes in acidity that facilitate the growth of upper intestinal bacterial in the gut, the effect of the number of drugs used could reflect either severe health conditions that impact microbiome composition or a bigger stress on the gut environment caused by multiple drug intake. In addition, we did not observe any change in the microbial richness with multi-drug use, suggesting that there is not a clear depletion or colonisation of bacteria.

We identified over 500 drug combinations in our cohorts. Even though we could not analyse specific drug combinations separately because of the limited numbers of each combination, we were able to show the importance of taking use of multiple drugs concurrently into account using two strategies: identifying associations in a multi-drug model and identifying associations in a single-drug models. As depicted in Fig. 2, we identified large differences in the number of associated taxa and pathways and the number of different drugs in the single-drug and multi-drug strategies. This demonstrates the added value of studying these interactions in patient groups where polypharmacy and comorbidities are common.

In the multi-drug meta-analysis, we identified that usage of PPI, laxatives and antibiotics had the largest effect on the gut microbiome composition. These three medication categories have different targets. Antibiotics directly target bacteria by inhibiting bacterial growth, while laxatives and PPIs have an impact on the host. A recent study, however, demonstrated that chemical compounds present in common medications can exhibit inhibitory effects on bacterial species[11]. In the case of PPIs, the impact on the gut microbial composition has been suggested to be a consequence of the combination of two mechanisms: indirect impact mediated by the changes in the gastrointestinal pH, which promotes the growth of typically oral bacteria, and a direct effect via inhibition of certain commensal gut bacteria that include *Dorea* and *Ruminococcus* species[7,11,25].

In our cohort, 30 participants were using, or had used, antibiotics in the 3 months prior to faecal sampling. Despite the limited number of users, we showed a decreased relative abundance of *Bifidobacterium* species in recent antibiotics users in the general population that is consistent with what has been described previously[4]. The decrease of *Bifidobacterium* abundances has also been shown in in vitro studies, where multiple antibiotic chemical components impact the growth of these bacteria[11].

A confounding factor in the study of the interaction between laxatives and gut microbiota is the difference in the intestinal transit time in patients using this medication due to diarrhoea or constipation. For example, increased abundances of *Bacteroides* species have been described in individuals experiencing a fast transit time[26]. This signature, however, has also been observed in

mice exposed to the laxative polyethylene glycol (PEG). While there is no evidence for a direct effect of this chemical compound on the inhibition of bacterial growth, experiments in mice suggest that microbial changes are indirect consequences of the disruption of the gut osmolality[27], and these changes seem to persist even weeks after the PEG administration. However, the long-term effect in humans has not yet been described. In addition, we found that the use of laxatives was associated with a higher relative abundance of the *Alistipes* genus. Interestingly, this genus has also been shown to be decreased in children with chronic functional constipation and to be resistant to bile acids[28], while another study identified a deficiency of bile acids in a subset of patients with IBS of the subtype with constipation[29]. These results may suggest a role for *Alistipes* in the pathogenesis of constipation.

We also identified an increase of *Methanobrevibacter smithii* in oral steroid users. This species has been associated with obesity and an increase in BMI in both rats and humans[30,31]. Pathways involved in methanogenesis were also increased in oral steroid users. However, these pathways were linked to the abundance of *Methanobrevibacter smithii*, therefore these functional changes are probably consequence of the increased abundance of this archaeon. It is believed that the methane produced by these species facilitates the digestion of polyfructose and thereby plays a role in caloric harvest[30,31]. This could potentially explain the weight gain frequently observed in oral steroid users[32]. In our study this effect was evident in patients with IBD, who were the cohort with the largest number of oral steroid users.

Species highly prevalent in the oral microbiome, like *Streptococcus parasanguinis*, are especially characteristic of the gut microbiome of PPI users, which is in agreement with a previously published study[8]. Correlated with this increase, specific AR mechanisms such as macrolide resistance also appear to be more abundant in faecal samples from PPI users. Previous studies have shown a synergistic effect of macrolides and PPIs, as indicated by the increased success rate of eradication therapy for *Helicobacter pylori* in patients receiving macrolides and PPIs versus macrolides alone, and this effect does not appear to be pH-dependent in vitro. The macrolide clarithromycin also inhibits the metabolism of the PPI omeprazole[33,34]. Moreover, we also observed an increase in microbial functions characteristic of the oral bacteria, such as carbohydrate degradation pathways, and an increase in pathways involved in L-arginine biosynthesis. Interestingly, one previous study has shown an important role for L-arginine in the bioavailability of the PPI omeprazole, where L-arginine increases the stability and solubility of omeprazole[35].

Our results showed an important role for *Escherichia coli* species in the gut microbiota of metformin users. Even though we could not identify any taxa associated with metformin use, we did identify an increased predicted metabolic potential of this species. Two recent studies exploring the impact of metformin on the gut microbiota showed significant changes in the bacterial composition and metabolic potential[9,10]. Although both studies identify a significant enrichment of *Escherichia coli* in the faecal samples of metformin users, direct causality could not be established in in vitro experiments. In our meta-analysis, this trend was also observed, but did not reach significance after multiple testing corrections. This can be partially explained by the fact that this species is already enriched in the faecal microbiota of patients with IBD. Consistent with previous studies, we observed changes in lipopolysaccharide and carbohydrate metabolism. More detailed analyses showed an enrichment in *Escherichia coli*-annotated pathways and gene families. However, this could partially be due the overrepresentation of this species in the current databases. Moreover, we replicated the in vitro finding that the AR protein *emrE* was increased in metformin users in the general

population (two-sided Wilcoxon test, FDR = 0.011)[11], indicating that non-antibiotic drugs can also influence the resistome profiles.

Although an interaction between acetaminophen (paracetamol) and the gut microbiota has been described[36], we could not replicate this association in our study. In line with our results, an in vitro study by Maier et al. showed that the administration of acetaminophen did not have a negative impact on bacterial growth of 40 common gut species[11]. Therefore, the inclusion of metabolomic measurements together with host genetics is needed to identify the indirect effects of the microbe–drug interactions.

The complex interaction between the use of medication, the gut microbiota and confounding factors poses several limitations in our study. Firstly, the cross-sectional nature of this study cannot identify causality in the observed associations. Second, the use of medication by itself is indicative of changes in the host's health condition that may also be accompanied by changes in lifestyle, and both are known to influence the microbiome composition in the gut. Third, due to the wide range of disorders for which the commonly used medications described in this manuscript are prescribed, it is difficult to establish a direct relation between medication use and its confounders. For example, PPIs are indicated for treating gastroesophageal reflux, but they are also prescribed for disorders like bloating or co-administered with NSAIDs to prevent ulcers. Moreover, for drugs sold over-the-counter, the indication is usually unclear. On the other hand, when drugs are commonly prescribed for a unique indication, such as metformin for type-2 diabetes, it becomes difficult to distinguish between the impact of the disease on the gut microbiota and the effect of the medication use. Fourth, patients using multiple different drugs could be less healthy. Ideally, to pinpoint the causality of our observed associations, prospective studies are needed that look at metagenomes from stool samples taken at multiple time points before and after the start of specific drugs. To disentangle these complex relations, the combination of longitudinal studies (from pre-treatment to wash-out period) with in vitro experiments can be a good approach.

Metagenomic sequencing studies provide insight into the associations between the use of medication and the changes in the microbial population in the gut, which may be related to pharmacological mechanisms and side effects. The integration of multiple host and microbial measurements, however, is needed to completely understand the complexity of the pharmacomicrobiomics interactions. For example, faecal metatranscriptomics experiments will provide a better understanding of bacterial dynamics and their functional implications, while metabolic profiling can reveal important host–microbiota interactions that affect drug metabolism.

Despite these limitations, our study of microbiome and medication use shows consistent associations between the functions and composition of the faecal microbiome and the intake of medication. We further show that the use of multiple drugs is associated with overall gut microbiome composition, either as a result of the drugs themselves or as a proxy for the underlying diseases. It is therefore worth correcting for multiple drug use in future gut microbiome studies in both the general population and GI disease cohorts.

Together our results contribute to the current knowledge of drug–microbiome interactions in a clinical context and provide a basis for further investigations of pharmacomicrobiomics and the potential gut-microbiota-driven side effects of currently prescribed drugs.

## Methods
**Cohort description.** For this study we used three independent Dutch cohorts: (1) a general population cohort, LifeLines-DEEP, consisting of 1539 individuals; (2) 544 patients with IBD from the 1000 IBD cohort of the University Medical Center of

Groningen (UMCG); and (3) an IBS case–control cohort with 313 participants from Maastricht University Medical Center + (MUMC +)[37–39]. The LifeLines-DEEP cohort consists of volunteers from the general population from whom faecal samples have been collected. Additionally, these participants filled in questionnaires about their (gastrointestinal) health, lifestyle and medication use. Participants from this cohort have a mean age of 44.8 with a standard deviation of 13.7, 58% are female and the mean BMI is 25.3 with a standard deviation of 4.2. The IBD cohort consists of patients for whom the diagnosis IBD has been made via standard radiological, endoscopic and histopathological evaluation by their treating physicians. Information about their medication use is derived from their medical records. For these IBD patients the mean age is 42.8 with a standard deviation of 14.8, 59% are female and the mean BMI is 25.5 with a standard deviation of 5.0. The IBS cohort consists of patients with IBS for whom a diagnosis has been made after extensively ruling out other diagnoses that might explain their GI complaints, usually also including an endoscopy. Furthermore, this cohort contains age- and sex-matched controls. Their medication use is derived from their medical records. In this cohort the mean age is 45.4 with a standard deviation of 17.7, 65% are female and the mean BMI is 24.6 with a standard deviation of 4.0. Each medication was classified into a drug category based on its indication following the Anatomical Therapeutic Chemical code (ATC-code) database and its working mechanism reviewed by medical doctors (Supplementary Data 1).

**Ethical approval.** All participants signed an informed consent form prior to sample collection. Institutional ethics review board (IRB) approval was available for all three cohorts: Lifelines DEEP (ref. M12.113965) and 1000 IBD (IRB-number 2008.338) cohort were approved by the UMCG IRB and the Maastricht IBS (ref. MEC 08-2-066.7/pl) cohort was approved by the MUMC + IRB.

**Sample collection and sequencing.** Each participant was asked to collect and freeze the faecal samples at home. Samples were then picked up and transported on dry ice and stored at –80°C. DNA extraction was done using AllPrep DNA/RNA Mini kit (Qiagen; cat. #80204) combined with mechanical lysis. Metagenomic shotgun sequencing was performed at the Broad Institute (Boston, Massachusetts, USA) using Illumina HiSeq platform, and low-quality reads were filtered out in the sequencing facility.

**Faecal sampling collection and metagenomic profiling.** Metagenomic reads mapping to the human genome or aligning to Illumina adapters were removed using *KneadData* (v0.4.6.1). After quality control of the sequenced reads, the microbial taxonomic and functional profiles were determined using *MetaPhlAn2* (v 2.2)[40] and *HUMAnN2* (v 0.10.0)[41], respectively. The Uniref90 and Chocophlan databases were used as a reference for microbial gene identification. Resistome characterisation was performed using *ShortBRED*[42] and the pre-calculated antibiotics database provided with the tool (accession July 2017). Default parameters for these tools were used.

**Sample exclusions and diversity measurements.** In the IBD cohort, 67 patients with stomas, pouches or short bowel syndrome were excluded. Furthermore, samples with a sequencing depth <10 million reads were removed ($n = 30$, 22 samples from the IBD cohort and 8 samples from the Maastricht IBS cohort). After filtering, 1883 samples remained for the analyses.

Microbial diversities and dissimilarities were computed using taxonomical *end-points* defined as the lower and non-redundant taxonomic level for each branch of the phylogenetic tree. Bray–Curtis distances and Shannon index were calculated using the functions *vegdist (method = "bray")* and *diversity (index = "shannon")* implemented in the R package *vegan* (v. 2.4-1). Microbial taxa were removed if they were redundant, absent in at least 90% of the samples in each cohort, or if the mean abundance of a taxon was <0.01% in each cohort. The remaining taxa values were normalised using the square root arcsine transformation. Microbial pathways were transformed to relative abundance and considered for analyses if they were present in >10% of the samples in each cohort. Finally, pathway abundances were log transformed, keeping the zero values. In total, 194 taxonomical end-points and 321 pathways were evaluated.

**Associations with microbial community measurements.** The association between each drug and bacterial richness (Shannon Index) was evaluated by performing two-sided Wilcoxon signed-rank tests between users and non-users. The impacts of medication categories on overall microbial composition (Bray-Curtis dissimilarities) were estimated via a PERMANOVA test with 10,000 permutations as implemented in the *adonis* function the R package *vegan*. In addition, the association between the number of administered drugs per participant, microbial diversity and composition were tested. Significance levels were adjusted for multiple testing using the Benjamini–Hochberg method.

**Drug–microbiome associations.** Medication categories were considered for analysis if they were used by at least 5 participants in each cohort. Drug associations with microbial features were initially evaluated per cohort using multivariable linear regression models adjusting for age, sex and BMI of each participant and the

sequencing depth of the sample. In the IBS cohort and the general population cohort, the IBS status was considered (yes/no) as covariate. Likewise, in cohort of patients with IBD, disease sub-phenotype was also added as a variable in the models (Crohn's disease, ulcerative colitis or IBD type unclassified).

Due to the multiple drug combinations it was not possible to estimate the effect of drug co-administration. However, to correct for this potential effect, two models were constructed:

(i) Association between individual taxa or pathways and specific drug types, adjusting for the general host factors: age, sex, BMI, sequencing depth and diseases. Represented as:
lm (feature ~ intercept + Drug + Age + Sex + BMI + Seq.depth + Disease (IBS/CD/UC/IBDU))

(ii) Association between individual taxa or pathways and specific drug types adjusted for host factors (age, sex, BMI, and sequencing depth), disease and the effect of the other drugs available in our metadata. Represented as:

lm (feature ~ intercept + Age + Sex + BMI + Seq.depth + Disease (IBS/CD/UC/IBDU) + Drug1 + … + Drug N)

We run both models in each cohort separately. For each of two models we extracted standard error and regression coefficient of the drug variable from single cohort analysis, and combined these in a meta-analysis framework. Specifically, the inverse variance weighted meta-analysis approach was used to calculate the combined meta z-scores and the corresponding meta p-values[43]. P-values were further adjusted for multiple testing using the Benjamini-Hochberg calculation as implemented in the *p.adjust* function in R. Associations with an FDR < 0.1 were further tested for heterogeneity using the Cochran's Q-test as implemented in the function *metagen* (*meta* R package (v. 4.8-4)). Finally, associations were considered to be significant if the meta-analysis multiple testing adjusted p-values were <0.05 (FDR < 0.05) and the heterogeneity *p*-value > 0.05.

**Resistome analysis.** AR gene abundances were calculated as the mean value of the normalised read counts of each marker representing a gene. AR genes present in < 10% of the participants were excluded from further analyses. Drug users were compared to participants not using any drugs in each cohort separately by performing a two-sided Wilcoxon test.

**Taxonomic contribution to metabolic pathways.** Pathways that were shown to be associated with medication use in the multi-drug meta-analysis were further investigated. To estimate the bacterial contribution to each pathway, we calculated the species-level stratified abundances using the HUMAnN2 pipeline. Gene families were also extracted using the *humann2_unpack_pathways* script. Values were transformed to relative abundance and log-transformed as described above. For each medication category associated with changes in the metabolic potential of the gut microbiota, the differential abundances in the stratified pathways and gene families were tested using the two-sided Wilcoxon signed-rank test. Significant levels were adjusted for multiple testing applying the Benjamini–Hochberg correction.

**Individual medication and dosage-dependent effects.** Statistically significant drug–microbiome associations were further assessed for the differential influence of drug types within the same category and the prescription dosages. Medication subtypes were analysed if they were present in at least 5 participants. To evaluate the effect of each medication subtype, the abundance of the associated microbial features was compared between users of a drug subtype and participants not using drugs belonging to the same category, for example a comparison of tetracycline users to participants not using antibiotics. Due to the distribution of the data referring to medication doses (Supplementary Data 12), samples were grouped into two categories: "high dose" and "low dose" of each particular drug. For PPIs this threshold was set to a minimum of 40 mg/day for the high dosage group and for metformin this minimum was set at 1000 mg/day. Users of laxatives, alpha-blockers, SSRI antidepressants or antibiotics in our cohort reported similar prescription patterns or the subtypes within this medication categories showed major differences in dosages. Therefore, we were unable to analyse dosages in these medication categories. Differences between groups were tested using a non-parametric *t*-test (two-sided Wilcoxon-test).

**Reporting summary.** Further information on research design is available in the Nature Research Reporting Summary linked to this article.

## Data availability

All relevant data supporting the key findings of this study are available within the article and its Supplementary Information files. Data underlying Figs. 1–4 are provided as a Source Data file. Other data are available from the corresponding author upon reasonable requests. The raw metagenomics sequencing reads and host-phenotype metadata used for the analysis presented in this study are available from the European Genome-phenome Archive data repository: 1000 IBD cohort [https://www.ebi.ac.uk/ega/datasets/EGAD00001004194], LifeLines Deep cohort [https://www.ebi.ac.uk/datasets/EGAD00001001991], Maastricht IBS cohort [https://www.ebi.ac.uk/datasets/EGAD00001002668]. Due to participant confidentiality, the datasets are available upon request to the University Medical Center of Groningen (UMCG),

LifeLines and Maastricht University Medical Center, respectively. This includes the submission of a letter of intent to the corresponding data access committees (1000 IBD Data Access Committee UMCG, LifeLines Data Acces Committee and Maastricht Irritable Bowel Syndrome Metagenomics Data Access Committee). Data access is subject to local rules and regulations.

## Code availability

For this study the following publicly available software was used: kneadData v0.4.6.1, MetaPhlAn2 v 2.2, HUMAnN2 v 0.10.0, ShortBRED v 1, R v 3.3.2; metagen (R package) v. 4.8-4, and vegan (R package) v. 2.4-1. Codes used for generating the microbial profiles are publicly available at: [https://github.com/WeersmaLabIBD/Microbiome/blob/master/Protocol_metagenomic_pipeline.md]. Codes used for generating the resistome profiles are publicly available at: [https://github.com/WeersmaLabIBD/Microbiome/blob/master/Protocol_antibiotic_resistance_genes_ShortBred.md]. Code used for the statistical analyses is publicly available at: [https://github.com/GRONINGEN-MICROBIOME-CENTRE/Groningen-Microbiome/tree/master/Projects/Medication_metanalysis].

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

## Acknowledgements

We would like to thank all participants from the three cohorts used in this study for providing their phenotypes and stool samples: LifeLines DEEP, the UMCG IBD cohort and the Maastricht University Medical Center IBS cohort. Furthermore, we thank B.H. Jansen for logistic and laboratory support and the IBD nurses at the UMCG IBD clinic, M.A.Y. Klaassen and L. Bolte, for patient inclusion and collection of the samples. We also thank K. McIntyre, who is employed by the Department of Genetics, UMCG, for English and scientific editing services. R.K.W., A.Z. and J.F. are supported by VIDI grants (016.136.308, 016.178.056 and 864.13.013) from the Netherlands Organization for Scientific Research (NWO). R.K.W. is further supported by a Diagnostics Grant from the Dutch Digestive Foundation (D16-14). A.Z. holds a Rosalind Franklin fellowship from the University of Groningen. Sequencing of the control cohort was funded by a grant from the Netherlands' Top Institute Food and Nutrition GH001 to C.W., who is further supported by an ERC advanced grant (ERC-671274) and a Spinoza award (NWO SPI 92-266). A.Z. is supported by an ERC starting grant (ERC-715772). J.F. and A.Z. are supported by a CardioVasculair Onderzoek Nederland grant (CVON 2012-03). Z.M. holds a Niels Stensen fellowship (from Amsterdam, the Netherlands).

## Author contributions

R.K.W. designed the study. A.V.V., V.C., A.R.B., T.S., F.I and Z.M. gathered and prepared the data. A.V.V. and V.C. analysed the data. A.V.V. and V.C. wrote the manuscript. S.S. and A.K. provided statistical advice. S.S., F.I., T.S., Z.M., D.M.A.E.J., A.A.M.M., A.R.B., J.F., C.W., A.K., A.Z. and R.K.W. were involved in sample collection, data generation and critically reviewing the manuscript.

## Competing interests

The authors declare no competing interests.
