## [Peer Review File · Nature Communications]

Reviewers' comments:

Reviewer #1 (Remarks to the Author):

Vich Vila et. al. examined the impact of 41 commonly-used medications on taxonomic structures and metabolic potential in the gut microbiome using data drawn from three large, deeply sequenced population-based cohorts. Performing a meta-analysis of studies, these authors found that 18 medications were associated with various microbial features (taxa and metabolic pathways) and that an increasing number of medications was linked to variation in beta diversity. Key strengths of this study include the size of the dataset, a large proportion of participants using medications, and mutual adjustment for other drugs. Overall, this study generates many hypotheses and reinforces the role of pharmaceuticals in driving interindividual variation seen in the gut microbiome. However, given the cross-sectional and correlative nature of the analyses, with limited reproducibility, there are few new and meaningful conclusions to be drawn.

Major points

1. The authors appropriately acknowledge in the discussion section that it is hard to eliminate confounding by indication. For example, if there are taxonomic differences observed among those taking antiplatelet drugs, could this be the result of having coronary artery disease instead of the medications? However, the authors should go further to acknowledge other residual confounders. Each medication, and subsequently each disease, is linked to other important lifestyle and dietary information, which are notably not included here.
2. I wonder if the authors have more granular information about medication use. I would like to see associations according to dose or duration (for those which are significant). Also, for some classes of medicines (e.g., NSAIDs), there are important differences within class that should be examined. For example, it is unclear where aspirin is included in this category. The biological effects of aspirin are likely quite different from other NSAIDs. Similarly, I would be curious to see individual effects of antibiotics, rather than all antibiotics lumped together.
3. Conspicuously absent from the analyses are analyses pertaining to IBD-associated medications. Although these would not be included in a meta-analysis, it would be important and interesting to show associations between anti-TNF therapy or 5-ASA compounds and composition/metagenomes.
4. The analyses presented leave the reader wanting to see links between changes in taxonomic data to changes in metagenomic data. The pipelines that the authors used give them optionality to see species-level contributions to gene families. Can they include this data as well? The results would be more compelling if streptococcal species, which are associated with PPI use, are indeed the dominant contributors to L-arginine synthesis.

Minor points

1. Some of the references do not seem to match up with those cited in text (i.e. reference 15)
2. Thyrox and ferrum should be listed by their generic names
3. Opiates and melatonin are misspelled
4. It would be nice to discuss the null acetaminophen results in the context of the PNAS acetaminophen study from 2009

Reviewer #2 (Remarks to the Author):

In this study, Vila et al. investigated the impact of medical drugs on the gut microbiome of three different study cohorts (population, IBD, and IBS cohort) using metagenomic data. The authors first performed a systematic characterization of drug co-administration in the three cohorts to subsequently search for associations between drug use and either compositional features or microbial pathway abundance or the presence of antibiotic resistance genes in the microbiome.

This work compiles a valuable and important resource for the research community. In particular the analysis of the compositional changes associated with the use of drugs and their combinations are carefully performed. The study however remains purely descriptive and interpretations of the obtained results are speculative. In particular, the interpretations of the functional analysis of metabolic pathways is superficial. E.g. the fact that genes of a particular metabolic pathway are enriched in metagenomic data under a condition of interest does generally not inform about the use of this pathway. This is in contrast to metatranscriptomics data. The authors should take this into account when discussing the results of bacterial pathway analysis.

Specific Comments:

1. Bacterial pathway analysis: Results of this analysis need to be discussed more thoughtfully throughout the manuscript:

E.g. line 131: it is unclear how it can be concluded from metagenomic data that "...steroids were associated with increased bacterial activity".

E.g. line 130: "essential pathways" can only be defined in a given ecological context. If the abundance of a pathway is reduced under given in vivo conditions, it is clearly not essential.

E.g. line 155-162: The authors can explain the PPI-association with increased abundance of genes involved in purine nucleoside degradation with the increased abundance of *Streptococcus* sp. From this results it could be concluded that PPI-use favors *Streptococcus* sp. rather than bacteria capable of purine nucleoside degradation (none of the other 26 bacterial genera encoding a purine nucleoside degradation pathway was enriched). This type of analysis should be performed and discussed for all associations of drug use with bacterial pathways to investigate whether it is rather a particular function or a genus/species that is favored by a given drug.

2. PPI, laxatives, and antibiotics had the biggest effect on microbiome composition. Whereas antibiotics directly impact bacterial growth, the other two drug types impact intestinal physiology. The authors should discuss this further and put it into the context of the in vitro study by Maier et al. that extensively investigated effects of non-antibacterial drug on microbial fitness.

3. The authors suggest that metformin use promotes *E. coli* activity. This raises two questions: i) It is unclear what *E. coli* activity is (see also comment 1). Are *E. coli* strains more abundant or are there genes of an *E. coli* specific pathway (e.g. colibactin biosynthesis) more abundant under metformin? ii) Functional pathway analysis resulting in *E. coli* specific findings is prone to be biased by the fact that annotations are best curated for this model organism. The authors should take this into account for their analysis and discussion.

4. The authors perform systematic analysis of drug co-administration and find that there is a strong correlation between steroid and beta sympathomimetic inhalers (first paragraph of results). In general, it does not become clear, how this data on drug co-administration is used in the subsequent association analysis. It seems that only the number of administered drugs has an impact on microbiome composition.

5. Given the resource-character of this work, raw sequencing data, including metadata, should be made freely accessible (without required permission) from one of the common databases.

Reviewers' comments:

Reviewer #1 (Remarks to the Author):

Vich Vila et. al. examined the impact of 41 commonly-used medications on taxonomic structures and metabolic potential in the gut microbiome using data drawn from three large, deeply sequenced population-based cohorts. Performing a meta-analysis of studies, these authors found that 18 medications were associated with various microbial features (taxa and metabolic pathways) and that an increasing number of medications was linked to variation in beta diversity. Key strengths of this study include the size of the dataset, a large proportion of participants using medications, and mutual adjustment for other drugs. Overall, this study generates many hypotheses and reinforces the role of pharmaceuticals in driving interindividual variation seen in the gut microbiome. However, given the cross-sectional and correlative nature of the analyses, with limited reproducibility, there are few new and meaningful conclusions to be drawn.

Major points

1. The authors appropriately acknowledge in the discussion section that it is hard to eliminate confounding by indication. For example, if there are taxonomic differences observed among those taking antiplatelet drugs, could this be the result of having coronary artery disease instead of the medications? However, the authors should go further to acknowledge other residual confounders. Each medication, and subsequently each disease, is linked to other important lifestyle and dietary information, which are notably not included here.

We thank the reviewer for bringing out this topic, which indeed was lacking in our discussion. As the reviewer has pointed out in this comment, the use of medication can be indicative of health conditions, and therefore, it becomes challenging to study the relation between changes in the gut microbiota composition and the medication usage. In addition, usage of medication is commonly complemented with changes in the lifestyle (for example, diet) that can also have an impact on the microbial composition in the gut. This complex relation is also relevant in the study of host-disorders. While most of the studies typically consider the use of antibiotics as an excluding or correcting factor in their analyses, the effect of other commonly used medications is still underestimated.

In the current version of the manuscript we have summarised additional potential confounding effects in the discussion. In addition, and following the suggestions of Reviewer 2, we have expanded the discussion on the identified associations and their relation with previous published findings (**see answers 2.2 and 2.3**) .

Lines 343 - 358

“The complex interaction between the use of medication, the gut microbiota and confounding factor, poses several limitations in our study. Firstly, the cross-sectional nature of this study cannot identify causality in the observed associations. Second, the use of medication by itself is indicative of changes in the health condition of the host, that may also be accompanied by changes in lifestyle, which are both known to influence the microbiome composition in the gut. Third, due to the wide range of disorders that the commonly used medications described in this manuscript are used for, it is difficult to establish a direct relation between medication use and its confounders. For example, PPIs are indicated for treating gastroesophageal reflux (GERD), but are also prescribed for disorders like bloating or co-administered with NSAIDs to prevent ulcers. Moreover, for drugs sold over-the-counter the indication is usually unclear. On the other hand, when drugs are commonly prescribed for a unique indication, such as metformin for type 2 diabetes, it becomes difficult to distinguish between disease impact on the gut microbiota and the effect of the medication use. Fourth, patients using multiple different drugs could be less healthy. Ideally, prospective studies with metagenomes from stool samples are needed at multiple time points, before and after start of certain drugs, to pinpoint the causality of our observed associations. To disentangle these complex relations, the combination of longitudinal studies (from pre-treatment to wash-out period) with in-vitro experiments can be a good approach.”

Rationale:

Although the effect of diet and lifestyle are relevant contributors of the microbial composition, they explain a relatively small proportion of the interindividual variation which we and others have shown previously¹⁻³. In our current analysis, we controlled for the effect of age, sex, sequencing depth and body mass index (BMI). The latest is known to be related with diet and lifestyle, and therefore, we expect to capture part of this effect when correcting for BMI.

In this study we identified 6 drugs to be associated with pathways or taxonomy when taking the use of other drugs into account. An important consideration is that we can divide these identified drugs into two groups, namely first the drugs which are only prescribed in one

disease, for example metformin in type 2 diabetes (T2D). Therefore, to assess the real effect of the drug we would need to compare patients with T2D not using metformin with those using it. Unfortunately, the number of T2D patients not using metformin is very limited in our cohort, since metformin is the first choice of drug in this patient group in the Netherlands.

The second group of drugs are those who are prescribed for numerous indications, this could either be for numerous diseases or for multiple symptoms, for example the proton-pump inhibitors (PPIs). These show one of the strongest associations with the microbiota in our study, are used for various indications: gastroesophageal reflux (GERD), but also in the context of bloating or the prevention of ulcers in therapies involving other drugs like NSAIDs. Another fact to consider is that these drugs can be sold over the counter in the Netherlands. In this case, and besides this heterogeneity and multiple confounding factors, the earlier findings using sequencing data were later replicated in longitudinal and in-vivo studies⁴⁻⁸, showing that although the known limitation of cross-sectional sequencing studies, this kind of research can be useful in the discovery of drug-microbiota associations.

2. I wonder if the authors have more granular information about medication use. I would like to see associations according to dose or duration (for those which are significant). Also, for some classes of medicines (e.g. NSAIDs), there are important differences within class that should be examined. For example, it is unclear where aspirin is included in this category. The biological effects of aspirin are likely quite different from other NSAIDs. Similarly, I would be curious to see individual effects of antibiotics, rather than all antibiotics lumped together.

We agree with the reviewer and now we have added more details on the specificities of the medication categories and dosages, however, data regarding the duration was not available in our cohorts. Due to the size of the table, this information can now be found in the updated Supplementary table 1 for convenience.

To compile this information, we had to go back to the original data source, which included questionnaires in the case of the population cohort and medical records in the case of the patients cohorts (both IBD and IBS). For that, and due to privacy restrictions, we had to request consent to the participants to access the data and ask medical doctors to retrieve that information from the electronic patient files. Unfortunately, this process has delayed our reply to the reviewers.

Medication classification

Following the comment of the reviewer we revised the classification of medication subtypes and added this information in the Supplementary tables 1. In brief, our categorization follows the ATC database classification, classifying each drug based on their indications. Moreover, we reviewed these groups creating sub-categories based on the chemical structure or working mechanisms. For example, antidepressant drugs were divided into 3 groups: SSRI-antidepressants, tricyclic antidepressants and a general category that represented the remaining antidepressant drugs. Regarding aspirin use, we did not categorize it in the NSAID group. In the Netherlands, this drug is prescribed as a platelet aggregation inhibitor which leads to lower dosages of the drug used in the Netherlands (80 milligrams/day) (higher doses of aspirin are needed to function as a painkiller and those are not prescribed in the Netherlands). Regarding the question on antibiotic usage, in our cohort there were 30 antibiotic users, for which the most prevalent ones were tetracyclines (n=9), penicillines (n=7) and fluorquinolones (n=6) (Supplementary table 1).

Moreover, and as the reviewer suggested we investigated if there was a differential effect depending on specific medication type, e.g. comparing tetracycline users with non-antibiotic users and penicillin users with non-antibiotic users. Strikingly, we found that different PPIs showed a similar effect, while in the category of antibiotics the associations were mostly derived from tetracyclines users. The relatively low numbers of antibiotic users prevented us however to identify major differences between the different types of antibiotics. All results are now summarized in the Supplementary table 12.

To clarify how each medication was classified we have now added the following text in the method section in lines 381 - 385, as well as the detailed information on Supplementary table 1:

Cohort description

Drug usage was retrieved from questionnaires in the population cohort and from medical records in the IBD and IBS cohorts. Each medication was classified into categories based on its indication following the Anatomical Therapeutic Chemical code (ATC-code) database and its working mechanism reviewed by medical doctors (Supplementary Table 1)."

Medication dosage

Regarding medication dosage, we retrieved the dosages from participants for those microbiome-associated drugs of the multivariate analyses as suggested by Reviewer 1. This was the case for the drugs SSRI-antidepressants, alpha-blockers, antibiotics, laxatives, proton pump inhibitors and metformin. However, differences in doses could not be tested on the

antibiotics, alpha-blockers, SSRI-antidepressant or laxatives due to the standardized prescription dosages (almost all participants using the same dosages). In the case of PPIs and metformin users, since most of the participants were using comparable doses, users were separated into two categories “high dosages users” and “low dosages users”. For PPI users, dosages less than or equal to 20 mg/day were considered as a low dosage and higher than 20 mg/day was considered as a high dosage. For metformin this cutoff was set at less than 1000 mg/day for the low dosage users.

In total, 46 pathways associated with PPI use showed a dosage effect, however, no significant associations were observed in metformin users (FDR<0.05, Supplementary table 12).

The following text has been added in the main manuscript in lines 459 - 472t:

“Methods:

Individual medication and dosage-dependent effects

Statistically significant medication-microbiome associations were further assessed for the differential influence of drug types within the same category and the prescription dosages. Medication subtypes were analysed if they were present in at least 5 participants. To evaluate the effect of each medication subtype, the abundance of the associated microbial features was compared between users of a drug subtype and participants not using drugs belonging to the same category. An example of that is the comparison between tetracycline users to participants not using antibiotics. Due to the distribution of the data referring to medication doses (Supplementary table 12), samples were grouped into two categories: “high dose” and “low dose” of each particular drug. For PPI’s this threshold was set to a minimum of 40 mg/day for the high dosage group and for metformin this minimum was set at 1000mg/day. Users of laxatives, alpha-blockers, SSRI-antidepressants or antibiotics of our cohort, reported similar prescription patterns or the subtypes within this medication categories showed major differences in dosages. Therefore, we were unable to analyse dosages in these medication categories. Differences between groups were tested using non-parametric t-test (wilcoxon-test).

Results:

Lines 182 - 187 (PPIs)

(...)Different types of PPIs, namely omeprazole, esomeprazole and pantoprazole, exhibited similar effects on the gut microbiome. Additionally, of the 131 microbial pathways associated with PPI use, 46 pathways also showed dosage dependent effects (FDR<0.05). For example, participants using a higher dosage of PPIs (more or equal to 40 mg/day) showed a marked

decrease in a pathway involved in the biosynthesis of amino acids (PWY-724) when comparing to low dosage users (FDR = 0.00064, Supplementary table 12).

Lines 197-198 (Metformin)

"(...)Furthermore, we did not identify dosage dependent effects of metformin usage on the metformin-pathways associations."

Lines 212 - 215 (SSRI antidepressants)

This medication category included six different subtypes of drugs in which paroxetine represented 32% of the SSRI-users (supplementary table 1). Interestingly, and despite the low numbers, the increased abundance of Eubacterium ramulus was mainly observed in the paroxetine users (FDR = 0.003, Supplementary table XX).

3. Conspicuously absent from the analyses are analyses pertaining to IBD-associated medications. Although these would not be included in a meta-analysis, it would be important and interesting to show associations between anti-TNF therapy or 5-ASA compounds and composition/metagenomes.

We agree with the reviewer about the importance of highlighting IBD-specific medication. In the revised version we have added additional information on the IBD specific drugs anti-TNF α , mesalazines and thiopurines (Supplementary table 49). Only the abundance of an *Erysipelotrichaceae* species (FDR=0.047) was associated with mesalazines use, the other IBD specific drugs did not show any associations with microbial features. We have added the following text in the results and discussion section:

Results lines 86 - 89:

"Mesalazines (36%), thiopurines (33%) and anti-TNF inhibitors (25%) were present in the top 10 most-used drugs in the IBD cohort. Since thiopurines and anti-TNF inhibitors were solely used in the IBD cohort, these drugs were not included in multivariate analyses. In patients with IBS, the strongest correlation was the use of steroid inhalers with beta sympathomimetic inhalers (Pearson 0.81, p-value < 2e-16)."

Results lines 155 - 157:

"Conversely, the use of medication usually prescribed to treat IBD did not show strong associations with the microbial composition. Only the abundance of an Erysipelotrichaceae species was found to be slightly increased in mesalazine users (FDR=0.047) (Supplementary table 49)."

4. The analyses presented leave the reader wanting to see links between changes in taxonomic data to changes in metagenomic data. The pipelines that the authors used give them optionality to see species-level contributions to gene families. Can they include this data as well? The results would be more compelling if streptococcal species, which are associated with PPI use, are indeed the dominant contributors to L-arginine synthesis.

We thank the reviewer for the suggestion. In the revised version of the manuscript, we have added a deeper characterization of all drug-pathways associations by:

a) For each pathway associated to a medication category we explored which bacteria were contributing to the pathways abundance. Next, we compared these values between medication users and non-users. To do so, we retrieved the bacterial contribution of each pathway from HUMAnN2 default output. We filtered those pathways missing in more than 90% in each cohort and normalized as described in the method section of our manuscript. A non-parametric t-test (Wilcoxon-test) was performed to evaluate if the bacterial contribution of each pathway differed between users and non-users. Resulting p-values were adjusted for multiple testing using Benjamini-Hochberg calculation.

b) Investigating which gene families are implicated in each associated pathway: Gene families involved in specific pathways were retrieved using the *humann2_unpack_pathways* script which is provided with the software. Filtering and normalization were performed as described above and differential abundance between users and non-users were tested using Wilcoxon-test.

c) Updated figure 2 to make easier the interpretation of the data.

Regarding the associations with PPI use, the 125 microbial associated pathways were predicted from 201 known bacterial genomes. After filtering pathways which were at least present in 10% of the samples of each individual cohort, 3174 were considered for analysis. Consistently with the observations in the taxonomic analysis, *Streptococcus* species were the top contributors in the differential abundance of those pathways in all three cohorts. For example, 29 organisms were found to contribute to the pathway involved in the L-arginine biosynthesis via the acetyl cycle (MetaCyc ID ARGYSYNBSUB). Nonetheless, in PPI users the increased abundance of this pathway was mainly linked to *Streptococcus mutans* (FDR < 0.05 in the three cohorts). At the gene-family level, more than 30.000 Uniref90 gene families were identified to be involved in the 125 PPI-associated pathways. Our analysis at this level

revealed a similar pattern as previously described: being *Streptococcus* genes enriched in the gut microbiota of PPI users.

We repeated this type of analyses for each associated pathways and provided the data in the Supplementary table 50 to facilitate the interpretation of the results. In addition, we have added the following text (**see also question 2.1, 2.2 and 2.3**):

Results section (lines 170 - 180):

However, a closer look at the predicted microbial contribution and at the gene families involved in each pathway revealed that the enrichment of specific microbial mechanisms is likely to be explained by the changes observed in taxonomical composition. Purine deoxyribonucleoside degradation, a pathway used as a source of energy and carbon, was predicted from the genomes of more than 27 different bacterial genera (figure 2). The increase in this function in the gut microbiome of PPI users can be explained by an increased abundance of Streptococcus species (S.salivaris, S.parasanguinis and S.vestibularis) (FDR<0.05). Three pathways involved in L-arginine biosynthesis (MetaCyc ID: PWY-7400, ARGSYNBSUB and ARGSYN) were more abundant in the microbiome of PPI users. While several bacterial taxa, including Bifidobacterium and Ruminococcus species, were predicted to contribute to these pathways, only Streptococcus mutans pathways showed a significant enrichment (FDR<0.05, Wilcoxon-test). ”

Methods section (lines 411 - 419):

“Taxonomic contribution to metabolic pathways

Pathways that were shown to be associated with medication use in the multivariate meta-analysis were further investigated. To estimate the bacterial contribution to each pathway we calculated the species-level stratified abundances using the HUmann2 pipeline. Gene families were also extracted using the humann2_unpack_pathways script. Values were transformed to relative abundance and log-transformed as described above. For each medication category associated with changes in the metabolic potential of the gut microbiota, the differential abundances in the stratified pathways and gene families were tested using the Wilcoxon signed-rank test. Significant levels were adjusted for multiple testing applying the Benjamini-Hochberg correction.”

Minor points

1. Some of the references do not seem to match up with those cited in text (i.e. reference 15)

We thank the reviewer for noticing the mistake. The references have now been adjusted.

2. Thyrax and ferrum should be listed by their generic names

We have now listed them by their generic names: 'thyrax' has been replaced by 'levothyroxine' and 'ferrum' has been replaced by 'iron preparations'.

3. Opiates and melatonin are misspelled

We have corrected figure 1 and revised the spelling in the manuscript.

4. It would be nice to discuss the null acetaminophen results in the context of the PNAS acetaminophen study from 2009

We thank the reviewer for pointing us to this interesting study. We have added the reference in the discussion section at lines 338 - 342:

"Although an interaction between acetaminophen and the gut microbiota has been described 40, we could not replicate this association in our study. In line with our results, the in-vitro study of Maier et al. showed that the administration of acetaminophen did not have a negative impact on bacterial growth of 40 common gut species¹¹. Therefore, the inclusion of metabolomic measurements together with host genetics is needed in order to identify indirect effects of the microbe-drug interactions."

Reviewer #2 (Remarks to the Author):

In this study, Vila et al. investigated the impact of medical drugs on the gut microbiome of three different study cohorts (population, IBD, and IBS cohort) using metagenomic data. The authors first performed a systematic characterization of drug co-administration in the three cohorts to subsequently search for associations between drug use and either compositional features or microbial pathway abundance or the presence of antibiotic resistance genes in the microbiome.

This work compiles a valuable and important resource for the research community. In particular the analysis of the compositional changes associated with the use of drugs

and their combinations are carefully performed. The study however remains purely descriptive and interpretations of the obtained results are speculative. In particular, the interpretations of the functional analysis of metabolic pathways is superficial. E.g. the fact that genes of a particular metabolic pathway are enriched in metagenomic data under a condition of interest does generally not inform about the use of this pathway. This is in contrast to metatranscriptomics data. The authors should take this into account when discussing the results of bacterial pathway analysis.

Specific Comments:

1. Bacterial pathway analysis: Results of this analysis need to be discussed more thoughtfully throughout the manuscript:

E.g. line 131: it is unclear how it can be concluded from metagenomic data that “...steroids were associated with increased bacterial activity”.

We agree with the reviewer that the definition of bacterial activity is not clearly stated. Indeed, bacterial activity was not directly measured and the results previously linked to “bacterial activity” referred to the metabolic potential, calculated as the pathways inferred from metagenomic sequencing alignments. We have adjusted the sentence to make it more accurate. It now reads at line 129 - 131 :

“Interestingly, while the use of antibiotics was related with a decrease in microbial pathways such as amino-acid biosynthesis, the use of metformin were associated with increased bacterial metabolic potential”

In addition, in the current version of the discussion, we have now emphasized the limitations of metagenomic studies at lines 359 - 365:

“Metagenomic sequencing studies provide insight into the associations between the use of medication and the changes in the microbial population in the gut, which can explain pharmacological mechanisms and side effects. The integration of multiple host and microbial measurements, however, is needed to completely understand the complexity of the pharmacomicrobiomics interactions. For example, faecal metatranscriptomics experiments will bring a better understanding of bacterial dynamics and its functional implications, while metabolic profiling can reveal important host-microbiota interactions affecting the drug metabolisms”

E.g. line 130: “essential pathways” can only be defined in a given ecological context. If the abundance of a pathway is reduced under given in vivo conditions, it is clearly not essential.

We agree with the reviewer's correction. We have removed the expression “essential pathways” from the main text.

E.g. line 155-162: The authors can explain the PPI-association with increased abundance of genes involved in purine nucleoside degradation with the increased abundance of *Streptococcus* sp. From this results it could be concluded that PPI-use favors *Streptococcus* sp. rather than bacteria capable of purine nucleoside degradation (none of the other 26 bacterial genera encoding a purine nucleoside degradation pathway was enriched). This type of analysis should be performed and discussed for all associations of drug use with bacterial pathways to investigate whether it is rather a particular function or a genus/species that is favored by a given drug.

Following the reviewer's suggestion, we have now performed additional analyses on the bacterial metabolic potential (see also question 1.4). First, we looked at the species contribution in each of the associated pathways to determine if the changes at pathway levels were driven by specific bacteria and, second, we selected the gene families (Uniref90) involved in each of those pathways and look for enrichment between users and non-users of drug categories. We have added this information in the supplementary table and add a description of the results throughout the manuscript.

Regarding the changes associated with PPI use we saw that the enrichment of *Streptococcus* and *Veillonella* sp. is also reflected in the gene abundance and pathway contribution, suggesting the functional changes are mainly consequence of this enrichment. For example, the purine deoxyribonucleosides degradation pathway, which represents a mechanism described in *E.coli* in which purines are utilized as a source of carbon and energy, was detected in more than 20 different bacterial species in each cohort. However, when comparing PPI users vs non-users, the significant enrichment was only observed in those pathways predicted from *Streptococcus salivaris*, *S.parasanguinis* and *S.vestibularis* (FDR<0.05, Supplementary table 50).

Consistently with the changes observed in the gut microbiota of oral steroids users, the pathways significantly enriched were identified to belong to *Methanobrevibacter smithii*.

In the current version of the manuscript, we expanded the discussion on pathways finding, for PPI users see question 1.4, in the case of metformin we discuss the findings on question 2.3.

2. PPI, laxatives, and antibiotics had the biggest effect on microbiome composition. Whereas antibiotics directly impact bacterial growth, the other two drug types impact intestinal physiology. The authors should discuss this further and put it into the context of the in vitro study by Maier et al. that extensively investigated effects of non-antibacterial drug on microbial fitness.

We agree with the reviewer on the importance of the Maier et al study in the context of our research, therefore, we have now expanded the comparison between our results and their findings. Following the reviewer's suggestion, in the current version of the manuscript we have added the following text in the discussion section at lines 275 - 296:

"In the multivariate meta-analysis, we identified that usage of PPI, laxatives and antibiotics had the largest effect on the gut microbiome composition. These three medication categories have different targets: antibiotics directly target bacteria by inhibiting bacterial growth, and laxatives and proton-pump inhibitors have an impact on the host. A recent study, however, has demonstrated that chemical compounds present in common medication can exhibit inhibitory effect on bacterial species¹¹. In the case of proton-pump inhibitors, the impact on the gut microbial composition has been suggested to be the consequence of the combination of two mechanisms: indirect impact mediated by the changes in the intestinal pH, promoting the growth of typically oral bacteria, and a direct effect via the inhibition of certain commensal gut bacteria, including Dorea and Ruminococcus species^{7,11,27}.

In our cohort, a total of 30 participants were using, or had used, antibiotics 3 months previous to fecal sampling. Despite the limited number of users, we showed a decreased relative abundance of Bifidobacterium species in the general population cohort, consistent with what has been described previously⁴. The decrease of Bifidobacterium abundances has also been shown in in-vitro studies, where multiple antibiotic chemical components impact the growth of these bacteria¹¹.

A confounding factor in the study of the interaction between laxatives and gut microbiota is a difference in the intestinal transit time in patients using this medication, due to diarrhea or obstipation. For example, increased abundances of Bacteroides species have been described in individuals experiencing a fast transit time²⁸. This signature, however, has also been observed in mice exposed to the laxative polyethylene glycol (PEG). While there is no evidence of a direct effect of this chemical compound on the inhibition of bacterial growth,

experiments in mice suggest that microbial changes are indirect consequences of the disruption of the gut osmolality²⁹. These changes seem to persist even weeks after the PEG administration. However, the long-term effect in humans has not yet been described.

3. The authors suggest that metformin use promotes E. coli activity. This raises two questions: i) It is unclear what E. coli activity is (see also comment 1). Are E. coli strains more abundant or are there genes of an E. coli specific pathway (e.g. colibactin biosynthesis) more abundant under metformin? ii) Functional pathway analysis resulting in E. coli specific findings is prone to be biased by the fact that annotations are best curated for this model organism. The authors should take this into account for their analysis and discussion.

We agree with the reviewer that the statement was not accurate enough. In our meta-analysis, the abundance of *E.coli* was not significantly associated with the use of metformin. Although an increased abundance is observed in the population cohort, this effect was not replicated in the other two cohorts. Changes in the microbiome composition associated with gastrointestinal disorders (such as IBD or IBS) may explain the heterogeneity of this effect. Although the association between the increased abundance of *E.coli* and the use of metformin has been shown previously, in-vitro and in-silico experiments could not demonstrate the direct association.

In contrast, significant associations were observed between functional changes (predicted from metagenomic data) and metformin use.

We agree with the reviewer that, although we see an enrichment on certain genes and pathways predicted from *E.coli* strains pan-genomes, we cannot determine if the change in the metabolic potential in the gut microbiota of metformin users is solely due to *E.coli*. Therefore, metatranscriptomics and metabolomic approaches combined with culturomics and detailed analyses for specific species is the preferred approach to disentangle this association. We have now edited the manuscript accordingly to the previous explanation. We changed the results section title and added the following text at lines 197 - 206:

“Metformin use is associated to changes in enterobacteria metabolic potential

While changes in the abundance of Streptococcus, Coprococcus and Escherichia species were initially found to be enriched in metformin users, these associations were no longer significant when correcting for the use of other drug types. However, a suggestive association with Escherichia coli (p=0.0006, FDR=0.11) remained and, in the IBD cohort, the abundance

of *Streptococcus mutans* was slightly increased in participants using this drug (FDR=0.01) (Supplementary Tables 18).

Strikingly, the functional implications of metformin use were large even after correction for the use of other drugs, with 53 microbial pathways altered compared to the non-users. *Metformin use was associated with changes in the metabolic potential of the microbiome, in particular with increases in the butanoate production, quinone biosynthesis, sugar derivatives degradation and polymyxin resistance pathways. Interestingly, metagenomic pathways prediction and gene family analyses revealed that Enterobacteriaceae species, mainly Escherichia coli, were the major contributors to the functional changes associated with metformin use. Our data suggest that physiological changes induced by metformin can provide competitive advantage to enterobacterial species which could potentially have implications on health (Supplementary Table 19).*”

In addition, in the discussion section we have now added the following text at lines 324 - 342:

“Our results showed an important role for Escherichia coli species in the gut microbiota of metformin users. Even though we could not identify any taxa associated with metformin use, we did identify an increased predicted metabolic potential of this species. Two recent studies exploring the impact of metformin on the gut microbiota showed significant changes in the bacterial composition and metabolic potential^{9,10}. Although both studies identify a significant enrichment of Escherichia coli in the faecal samples of metformin users, direct causality could not be established in in-vitro experiments. In our meta-analysis, this trend was also observed. However, it did not reach the significance after multiple testing correction. This could partially be explained by the fact that this species is already enriched in the faecal microbiota of patients with IBD. Furthermore, the metabolic potential of the microbial ecosystem was altered in metformin users. Consistently with previous studies, changes were observed in the lipopolysaccharide and carbohydrates metabolism. More detailed analyses showed an enrichment in E.coli annotated pathways and gene-families, however, this could partially be due to the overrepresentation of this specie in the current databases. Overall, our results support the hypothesis that metformin has an indirect effect on the gut microbiota mediated by changes in the gut environment.”

4. The authors perform systematic analysis of drug co-administration and find that there is a strong correlation between steroid and beta sympathomimetic inhalers (first paragraph of results). In general, it does not become clear, how this data on drug co-administration is used in the subsequent association analysis. It seems that only the number of administered drugs has an impact on microbiome composition.

We want to thank the reviewer for pointing out that the methodology was not completely clear. Indeed, we report that certain medication groups, including the case of steroids and beta sympathomimetic inhalers, are frequently prescribed together in our cohort. We therefore considered investigating the potential interacting effect and/or stratifying participants based on combination of medications

In the first part of the result section we wanted to highlight the co-administration patterns of certain drugs in order to better characterize the drug usage in our cohorts and therefore, provide support for the interpretation of the results of the univariate drug-microbiota associations. In addition, we also focus in two broad microbiota metrics: richness (shannon index) and overall microbial composition (Bray Curtis dissimilarities matrix). We then showed that, in our cohort, the use of individual drugs do not alter the richness or the overall composition of the gut microbiota, with the exception of proton-pump inhibitors, which are significantly associated with changes in the Bray-Curtis dissimilarities. Due to the fact that the use of multiple drugs could mean the exposure of the intestinal microbiota to several external compounds but also a combination of effects in the host, we hypothesize that the number of administered drugs (number of drugs that a participant was taking at the time of fecal sampling collection) could have an impact on the richness and the overall composition. Interestingly, we show that the number of medications used by the host is associated with changes in the microbial composition which are probably also indicative of its health status.

In the second part of the study, we focus on the specific drug-microbiota association using two different models. Due to the multiple medication combination (more than 500), it was not possible to estimate co-administration effects. To correct for this (possible) effect, first, we considered the association between bacteria and each drug individually and then, we added all other medication categories in the same model to account for the confounding effect of co-administered drugs.

In the current version of the manuscript we have clarified the methods section at lines 421 - 428:

“Statistical analyses

Associations to microbial community measurements

The association between each drug and bacterial diversity (Shannon Index) was evaluated by performing wilcoxon signed-rank tests between users and non-users. The impact of medication categories on the microbial overall composition (Bray Curtis dissimilarities) were estimated using a PERMANOVA test with 10000 permutations as implemented in the adonis function of vegan R package. In addition, the association between number of administered drugs per participant, microbial diversity and composition were tested. Significance levels were adjusted for multiple testing with Benjamini Hochberg method.

Individual cohort associations

Drug associations with microbial features were initially evaluated per cohort using linear models. Due to the multiple medication combinations it was not possible to estimate the effect of drug co-administration, however, to correct for this potential effect, two models were constructed:

(i) Association between individual taxa or pathways and specific drug types, adjusting for the general host factors: age, sex, BMI and sequencing depth.

(ii) Association between individual taxa or pathways and specific drug types adjusted for host factors (age, sex, BMI and sequencing depth) and the effect of the other 40 drugs available in our metadata. Additional covariates were diagnosis (Crohn's disease, ulcerative colitis or inflammatory bowel disease type unclassified) in the IBD cohort and an IBS diagnosis in the Maastricht IBS cohort and in the general population."

5. Given the resource-character of this work, raw sequencing data, including metadata, should be made freely accessible (without required permission) from one of the common databases.

We fully agree with the reviewer and we are committed with the goal of making all research FAIR. All data used to carry out this research is available in the European Genome-phenome Archive. Due to the current privacy and IRB regulations at the time of sample collection, data can be freely shared with academic institution but with an obligated control step on the data access, consisting on reviewing applications before allowing access. Notice that the three datasets used in this study depend of three different institutions: LifeLines Deep population cohort (LifeLines), case control IBS cohort (Maastricht University Medical Centre) and the IBD cohort (University Medical Center of Groningen), therefore the access policy on the data depends on each of the owners.

Reference

1. Zhernakova, A. *et al.* Population-based metagenomics analysis reveals markers for gut microbiome composition and diversity. *Science* **352**, 565–569 (2016).
2. Falony, G. *et al.* Population-level analysis of gut microbiome variation. *Science* **352**, 560–564 (2016).
3. Vich Vila, A. *et al.* Gut microbiota composition and functional changes in inflammatory bowel disease and irritable bowel syndrome. *Sci. Transl. Med.* **10**, eaap8914 (2018).
4. Imhann, F. *et al.* The influence of proton pump inhibitors and other commonly used medication on the gut microbiota. *Gut Microbes* **8**, 351–358 (2017).
5. Hojo, M. *et al.* Gut Microbiota Composition Before and After Use of Proton Pump Inhibitors. *Dig. Dis. Sci.* **63**, 2940–2949 (2018).
6. Jackson, M. A. *et al.* Proton pump inhibitors alter the composition of the gut microbiota. *Gut* **65**, 749–756 (2016).
7. Reveles, K. R., Ryan, C. N., Chan, L., Cosimi, R. A. & Haynes, W. L. Proton pump inhibitor use associated with changes in gut microbiota composition. *Gut* **67**, 1369–1370 (2018).
8. Takagi, T. *et al.* The influence of long-term use of proton pump inhibitors on the gut microbiota: an age-sex-matched case-control study. *J. Clin. Biochem. Nutr.* **62**, 100–105 (2018).

REVIEWERS' COMMENTS:

Reviewer #1 (Remarks to the Author):

Thank you for your detailed response. The paper is an excellent contribution.

Andrew Chan

Reviewer #2 (Remarks to the Author):

The authors have addressed all my questions raised in the previous review and have made according clarifications in the results and methods parts.

I believe that the thorough analysis of this impressive amount of data and the provided supplementary tables will be a great resource for the scientific community aiming at future mechanistic studies to better understand the impact of drugs on the gut microbiome composition and functioning.

REVIEWERS' COMMENTS:

Reviewer #1 (Remarks to the Author):

Thank you for your detailed response. The paper is an excellent contribution.

Andrew Chan

Reviewer #2 (Remarks to the Author):

The authors have addressed all my questions raised in the previous review and have made according clarifications in the results and methods parts.

I believe that the thorough analysis of this impressive amount of data and the provided supplementary tables will be a great resource for the scientific community aiming at future mechanistic studies to better understand the impact of drugs on the gut microbiome composition and functioning

Response to reviewers' comments.

We thank reviewer 1 and 2 for their positive feedback and comments.